# Translational Read-Through Drugs (TRIDs) Are Able to Restore Protein Expression and Ciliogenesis in Fibroblasts of Patients with Retinitis Pigmentosa Caused by a Premature Termination Codon in *FAM161A*

**DOI:** 10.3390/ijms23073541

**Published:** 2022-03-24

**Authors:** Avigail Beryozkin, Ananya Samanta, Prakadeeswari Gopalakrishnan, Samer Khateb, Eyal Banin, Dror Sharon, Kerstin Nagel-Wolfrum

**Affiliations:** 1Hadassah Medical Center, Department of Ophthalmology, Faculty of Medicine, The Hebrew University of Jerusalem, Jerusalem 91120, Israel; avigail.beryozkin@mail.huji.ac.il (A.B.); prakadee.gopalakrish@mail.huji.ac.il (P.G.); samerkhateb@gmail.com (S.K.); banine@mail.huji.ac.il (E.B.); dror.sharon1@gmail.com (D.S.); 2Institute of Molecular Physiology, Johannes Gutenberg University of Mainz, 55122 Mainz, Germany; ananya.samanta85@gmail.com; 3Institute of Development Biology and Neurobiology, Johannes Gutenberg University of Mainz, 55122 Mainz, Germany

**Keywords:** FAM161A, retinitis pigmentosa, fibroblasts, ataluren, gentamicin, translational read-through drugs, TRIDs

## Abstract

Ataluren and Gentamicin are translational readthrough drugs (TRIDs) that induce premature termination codon (PTC) readthrough, resulting in the production of full-length proteins that usually harbor a single missense substitution. FAM161A is a ciliary protein which is expressed in photoreceptors, and pathogenic variants in this gene cause retinitis pigmentosa (RP). Applying TRIDs on fibroblasts from RP patients due to PTC in the *FAM161A* (p.Arg523*) gene may uncover whether TRIDs can restore expression, localization and function of this protein. Fibroblasts from six patients and five age-matched controls were starved prior to treatment with ataluren or gentamicin, and later FAM161A expression, ciliogenesis and cilia length were analyzed. In contrast to control cells, fibroblasts of patients did not express the FAM161A protein, showed a lower percentage of ciliated cells and grew shorter cilia after starvation. Ataluren and Gentamicin treatment were able to restore FAM161A expression, localization and co-localization with α-tubulin. Ciliogenesis and cilia length were restored following Ataluren treatment almost up to a level which was observed in control cells. Gentamicin was less efficient in ciliogenesis compared to Ataluren. Our results provide a proof-of-concept that PTCs in *FAM161A* can be effectively suppressed by Ataluren or Gentamicin, resulting in a full-length functional protein.

## 1. Introduction

Inherited retinal diseases (IRDs) can be caused by at least 300 different mutated genes (https://sph.uth.edu/retnet/, accessed on 25 December 2021) and can be inherited by autosomal recessive (AR), autosomal dominant (AD) and X-linked manners. The most common IRD is retinitis pigmentosa (RP), with a prevalence of 1:4500 in Europe and the USA [1,2,3,4], and much higher prevalence in the vicinity of Jerusalem (1:2100) [5]. There are dozens of different genes which can cause RP, one of which is *FAM161A* that was identified by us and others in 2010 [6,7], as a cause of ARRP when mutated. Since then, several studies were performed to uncover its function [8,9,10,11]. The *FAM161A* gene contains seven exons and encodes for several transcripts, two of which (with and without exon 4) were studied, but their exact difference in function or localization is still unclear [6,7]. The *FAM161A* gene is responsible for about 10% of nonsyndromic RP cases in the Jewish population in Israel and for more than 33% of nonsyndromic RP cases of North African Jewish ancestry [12]. So far, more than 100 RP patients with bi-allelic *FAM161A* mutations were identified in this population [12] due to only two founder mutations in exon 3: a frameshift (c.1355_6del) and a nonsense (c.1567C>T, p.R523*).

In the murine retina, Fam161a was localized in photoreceptor inner segments, connecting cilia, the synaptic regions of the outer and inner plexiform layers and ganglion cells [11,13]. Different studies showed that in the murine and human retinas, FAM161A is localized at the base of the connecting cilium, the basal body region and the adjacent centriole in photoreceptor cells [10,11,13]. Proteomics analyses show that FAM161A is part of the cytoskeleton fraction of the murine photoreceptor sensory cilium complex and a component of human centrosomes [14]. In addition, FAM161A was shown to be a part of microtubule-organizing centers, to have a role in stabilization of existing microtubules [11] and in the assembly of the primary cilium in cell cultures [10]. There are two different mouse models for *Fam161a*, displaying much shorter connecting cilia in photoreceptor cells before their degeneration, indicating that FAM161A has a role in stabilizing the photoreceptor connecting cilium as well [15,16]. FAM161A directly binds to microtubules and increases the acetylation of α-tubulin [10,11]. It was found to be a member of the Golgi-centrosomal interactome, a network of proteins interconnecting Golgi maintenance, intracellular transport and centrosome organization [9]. During cell cycle, FAM161A follows the centrosome through all stages of mitosis [9]. Thus, FAM161A is likely to be a multifunctional protein that is important for photoreceptor ciliary function as well as centrosomal functions.

In the current study, we focused on a founder nonsense mutation, *FAM161A*-c.1567C>T (p.R523*), that causes ARRP. Nonsense mutations, also known as pre-mature termination codons (PTC), affect translation, as it turns a coding triplet into a stop codon. PTCs cause either the degradation of mRNA by the nonsense mediated decay (NMD) surveillance or early termination of the translation process producing a short protein, which is often non-functional [17].

In the last few years, a gene-based therapy that is based on suppressing pathogenic nonsense mutations using translational read-through drugs (TRIDs) emerged. TRIDs are small molecules such as Ataluren (PTC124, under commercial name TranslarnaTM) and aminoglycoside antibiotics such as Gentamicin (aminoglycoside) which were described to induce PTC readthrough [17,18,19,20]. TRIDs reduce proofreading of codon–anticodon recognition in the ribosome, allowing the translation machinery to suppress different PTCs with different efficacy. This readthrough of a pathogenic nonsense mutation results in the synthesis of a full-length protein [18,21]. In most cases, PTC readthrough results in translation of a full-length protein with a different amino acid (compared to WT) at the location of the mutated codon [22,23]. Because the altered amino acid is equivalent to a missense change, the functionality of the protein might be variable, from a fully functional protein to non-functional or even toxic products [24]. To date, numerous studies demonstrated that the recovered proteins can be functional [25,26]. Moreover, in recessive and X-linked diseases, a relatively small amount of functional or partially functional protein might slow disease progression or result in a milder phenotype [27].

Ataluren was recently approved for treating a nonsense mutation causing Duchenne muscular dystrophy (DMD) in the USA and the EU, exhibiting a good safety profile and clinical benefit in the 6 min walk test in patients when taken orally, indicating its safety for systemic use [28]. Ataluren was also proved in an animal model to be safe and efficient for treatment of aniridia caused by a nonsense mutation in *PAX6*. In particular, as an eye drop formulation (START), Ataluren can be applied on eyes, and it is able to penetrate and be efficient to treat all the different layers of the eye in small rodents [29,30].

Here, we demonstrate the efficacy of Ataluren and Gentamicin on restoring protein expression and ciliogenesis in fibroblasts from six patients with RP caused by a homozygous *FAM161A* nonsense mutation (c.1567C>T). Our results provide a proof-of-concept that PTCs in *FAM161A* can be treated with TRIDs.

## 2. Materials and Methods

### 2.1. Patients

Skin biopsies were collected from the inner side of the arm of six patients with a confirmed homozygous nonsense mutation in *FAM161A* (c.1567C>T, p.Arg523*; Appendix A) and five age-matched healthy controls. All patients were diagnosed with RP following ERG testing and retinal imaging, and their clinical features were previously described [12]. None of the patients were on drugs or vitamins for managing RP. All participants in the study signed an informed consent form that adhered to the tenets of the declaration of Helsinki before drawing skin and/or blood samples. Ethical approval for this study was obtained from the Hadassah-Hebrew University Medical Center.

### 2.2. Cell Culture Treatment

Primary skin fibroblasts were grown in RPMI medium with 10% fetal bovine serum and antibiotics (penicillin and streptomycin) at 37 °C in 5% CO_2_. Early passage (up to 5th passage) of primary skin fibroblasts were plated in 9 cm cell culture plates and grown to 80% confluence. Ciliary growth was induced by Opti-MEM-Reduced Serum Medium (ThermoFisher Scientific, Waltham, MA, USA) for 48 h. Cells were treated with different TRIDs, which were added for 48 h to Opti-MEM media: Ataluren (Selleckchem, cat. Number S6003) and Gentamicin (Sigma-Aldrich, St. Louis, MO, USA) at a concentration previously described to promote a remarkable PTC read through (10 μg/mL Ataluren and 1 mg/mL Gentamicin, respectively [31]). DMSO (Dimethylsulfoxide) was used as a solvent and as a control for Ataluren.

A total of 300 cells from each patient or healthy control were examined (3 independent experiments, 100 cells each, Appendix A), and the following measurements were evaluated in treated and untreated cells from patients and healthy controls: cilia growth (PCTN2 ab—ABCAM (Cambridge, UK), ab99341 and Actub ab—Sigma, T6793), cilia length (as was measured by Image-J software, version 1.53k), formation of microtubules (by α-tubulin ab—Sigma, T9026) and FAM161A expression (using commercial antibody-rabbit anti-FAM161A, HPA032119, Sigma-Aldrich). Measurements of cilia which were lower than 1 µm were not included in the analyses.

### 2.3. Microscopy and Image Processing

The immunofluorescence staining of specimens was analyzed using a Leica DM6000B microscope using LAS-AF software (Leica, Bensheim, Germany). The following objectives were used: 506174: ∞/0.17/0, HCX PLANAPO, 60x/0.75 PH2 and 506182: ∞/0.17/0, HCX PL APO, 63x/1.32O/L PH3CS. Pictures were taken with the same intensities. Fluorescence images processing was performed with LAS-AF Leica imaging software and CanvasX/ImageJ/Fiji software [32,33]. For quantification of FAM161A, α-tubulin, Actub and PCN2 expression levels of the fluorescence intensities of each one of the proteins in 250 healthy untreated cells (50 from each participant) and 900 cells from patients (50 cells from each treatment, total of 150 cells from patient) were measured using ImageJ, and the corrected total cell fluorescence (CTCF) was calculated (arbitrary unit). CTCF = Integrated Density—(Area of selected cell × Mean Fluorescence of background readings) [34].

### 2.4. Statistical and Genetic Analyses

Statistical analyses were preformed using MS-Excel. The student’s *t*-test was performed to prove the significance of observed differences (unpaired, two-tailed, assuming equal variance). The significance levels were set when *p* < 0.05 (*), *p* < 0.01 (**), *p* < 0.001 (***).

The possible pathogenicity of the missense changes was evaluated using PolyPhen-2 (available online: http://genetics.bwh.harvard.edu/pph2/, accessed on 15 September 2021), MutationTaster (available online: http://www.mutationtaster.org/, accessed on 15 September 2021) and SIFT (available online: http://sift.jcvi.org/, accessed on 15 September 2021).

## 3. Results

The current study focuses on nonsense mutation c.1567C>T; p.Arg523* in the *FAM161A* gene, located in exon 3 (Appendix A) within the conserved UPF0564 domain that binds microtubules. Arginine at position 523 is highly conserved among various species including zebrafish (Table 1), indicating that this residue might be important for protein structure or function.

To evaluate the potential effect of different TRIDs on this mutation, we collected skin biopsies and established fibroblasts’ cell lines from six ARRP patients who are homozygous for FAM161A-p.R523*. Utilizing *in silico* tools, we analyzed all possible outcomes of treating p.R523* with TRIDs (Appendix A), and all alternative amino acids (except arginine) have physical properties that are different from the WT. It should be noted, however, that these tools are prone to non-accurate predictions and that no pathogenic missense variants were so far reported in *FAM161A*, indicating that the protein can tolerate such variants. Nonetheless, we extended our research and analyzed the effects of the recovered FAM161A protein expression in patient-derived fibroblasts following TRIDs treatment.

### 3.1. FAM161A Expression in Human Fibroblasts

We assessed the expression pattern of FAM161A in fibroblast cells by immunofluorescence labeling of FAM161A in fibroblast cells that were induced to ciliogenesis. The analysis revealed FAM161A expression only in the control cells (Figure 1A,Q) but not in *FAM161A* patient cells (Figure 1B). We subsequently performed double staining of FAM161A and α-tubulin, a molecular marker for microtubules in fibroblasts. We observed α-tubulin expression in 94% of the control cells (Appendix A). All control fibroblasts showed localization of α-tubulin in well-organized, fiber-like structures which constitute the microtubule network (Figure 1E). Staining of FAM161A (Figure 1I) overlapped with the microtubule labeling produced by anti-α-tubulin antibodies (Figure 1M), confirming the previously described interaction of FAM161A and microtubules [11]. In starved fibroblasts from *FAM161A* patients, immunolabeling of α-tubulin appeared in 85% of cells (compared to 94% in cells from controls—Appendix A), it was also significantly fainter and less structured when compared to control fibroblasts (Figure 1F,R). Extremely faint or even no FAM161A staining was observed in fibroblasts from FAM161A patients (Figure 1J,N).

### 3.2. FAM161A Expression Following TRIDs Treatment

In order to study the effect of TRIDs on FAM161A expression and localization, we treated fibroblast cells of *FAM161A* patients with either Gentamicin or Ataluren. Partial restoration of FAM161A protein expression was observed in fibroblast cells of *FAM161A* patients 48 h following TRID treatments (Figure 1C,D,K,L,Q). Quantification of the fluorescence intensities revealed a significant increase in the FAM161A expression in Gentamicin-treated and Ataluren-treated cells when compared to untreated patient cells (Figure 1Q). Expression of α-tubulin was observed in 91.1% of patients’ cells treated with Gentamicin and 93.6% of patients’ cells treated with Ataluren (Appendix A). In addition, treated cells showed fiber-like staining of α-tubulin (Figure 1G,H) and FAM161A (Figure 1K,L) which co-localized (Figure 1O,P), indicating restoration of both FAM161A expression and function. The effect of Ataluren was more pronounced than that of Gentamicin, as evidenced by the stripe-like orientation of α-tubulin, the co-localization of α-tubulin and FAM161A and the quantification of the fluorescence intensities (Figure 1O–Q; Appendix A).

### 3.3. Ciliogenesis Analysis in Fibroblasts

Since *FAM161A* encodes a ciliary protein, we analyzed ciliogenesis of primary cilia in *FAM161A* patient-derived fibroblasts compared to controls. After triggering ciliogenesis by starvation for 48 h, fibroblasts from five controls and six *FAM161A* patients were stained with anti-acetylated tubulin (Actub, green), a ciliary marker, anti-pericentrin2 (PCTN2, red), a basal body marker, and DAPI (blue) (Figure 2). Acetylated tubulin stainings were detectable in cilia and the cytoplasm of the fibroblasts; however, the staining was weaker in untreated and Gentamicin treated patient cells compared to healthy controls and Ataluren treated patient cells (Figure 2A–F). Pericentrin was detected at the base of cilia; but interestingly, quantification of immunofluorescence labeling of pericentrin revealed a reduced staining in untreated and Gentamicin treated patient cells, providing the first hint of reduced ciliogenesis in *FAM161A* patient cells (Figure 2A–E,G). The reduced acetylated tubulin labeling and pericentrin labeling could be restored by Ataluren treatment (Figure 2; Appendix A).

We subsequently compared the percentage of ciliated cells as well as cilia length between controls and patients’ cells (Figure 3A,B; Table 2). Control cells had a significantly higher percentage of ciliated cells when compared to *FAM161A* patients’ cells (1107 ciliated cells out of 1500 in control group [73.8%] vs. 617 out of 1800 cells originated from *FAM161A* patients [34.3%], *p* < 0.001; Figure 3A, Table 2). In addition, ciliated control cells had significantly longer cilia as compared to FAM161A mutant cells (average of 2.95 µm in control cells compared to 2.13 µm in *FAM161A* patient cells, *p* < 0.001; Figure 3B, Table 2). Performed *t*-tests did not show any significant differences in ciliogenesis or cilia length between young and old participants.

We subsequently examined the effect of Gentamicin or Ataluren treatment on patients’ fibroblast cell lines. Treatment with DMSO as a control did not show any effect on the percentage of ciliated cells or cilia length (Table 2 and Appendix A). Gentamicin treatment had no effect on the percentage of ciliated cells in treated versus untreated cells, both of healthy controls or patients (Table 2 and Appendix A). However, patient cells treated with Gentamicin had longer cilia compared to untreated cells (3.14 µm vs. 2.13 µm, *p* < 0.001, Table 2) and revealed strong expression of acetylated tubulin along the cilia and very blur staining of acetylated tubulin in other cellular areas (Figure 2D,F). On the other hand, Ataluren treatment of patient cells showed an effect on both parameters, resulting in a higher percentage of ciliated cells (66.6% compared to 34.4%, *p* < 0.001) (Figure 3A, Table 2 and Appendix A) and longer cilia (3.1 µm vs. 2.13, *p* < 0.001) (Figure 3B, Table 2). No significant difference between the numbers of ciliated cells was observed in healthy cells versus Ataluren-treated patient cells, demonstrating that Ataluren can restore ciliogenesis in *FAM161A* patient cells. We observed that Ataluren treatment in patients resulted in cilia being 2% longer compared to cilia of healthy control cells. So far, we do not know if an increase in cilia length in this low range might have physiological consequences in Ataluren-treated patients. Furthermore, we observed restored expression of acetylated tubulin along microtubules in Ataluren treated patient cells (Figure 2E,F), while as expected, Ataluren had no effect on control cells.

## 4. Discussion

Mutations in *FAM161A* are currently the most common cause for nonsyndromic ARRP in the Jewish population with more than 100 affected patients in the Israeli Jewish population due to only two pathogenic variants [12]. One of these pathogenic variants modifies the codon for arginine in position 523 into a PTC. In the current study, we aimed to provide a proof of concept that biallelic *FAM161A* patients harboring the c.1567C>T (p.Arg523*) mutation on at least one allele can potentially benefit from TRIDs therapy.

Previous studies showed FAM161A expression in different cell lines: hTERT-RPE-1, cos-7, HeLa and other ciliated cells [9,10,11]. Here we show for the first time that FAM161A is expressed in human fibroblast cells in which ciliogenesis can be triggered by starvation. Moreover, we demonstrate that in control cells FAM161A is co-localized with α-tubulin indicating its association with microtubules, as previously reported in other cells [10,11]. On the other hand, *FAM161A* patient-derived cells show a very weak FAM161A expression and modified α-tubulin expression. The observed weak expression might be either due to spontaneous read-through of the nonsense mutations, a phenomenon observed previously [35], or expression of a short nonfunctional protein. However, this low expression level is not enough for normal cell function, namely binding to microtubules and increasing the acetylation of α-tubulin [13].

FAM161A was reported previously to be a ciliary protein, and two mouse models were reported with significantly short cilia in photoreceptor cells, indicating that absence of this protein causes shorter cilia [15,16]. To this end, we show here that ciliogenesis (triggered by starvation) in *FAM161A* mutated fibroblast cells is reduced as compared to controls, and the generated cilia were significantly shorter. We assume that absence or extremely low expression of FAM161A affects cilia production and structure not only in mice photoreceptors but also in human fibroblasts, and potentially in the human retina as well.

In the present study, we used two different TRIDs, Gentamicin and Ataluren. Gentamicin was previously reported as efficient read-through treatment for different genetic disorders in cell cultures, animal models and humans [36]. It was also shown as an efficient drug for genetic eye diseases such as choroideremia and ocular coloboma in zebrafish models [37], and Usher syndrome in cellular models (HEK293T and fibroblasts) and mouse retinal explants [31,38]. Several side effects were reported, such as retinal-, nephron- and ototoxicity [26,38,39], which made Gentamicin not practical for long term use (systemically or directly administrated to the eye), though in several studies it was shown to be dose dependent [37,38]. We used a concentration that was previously reported efficient and not toxic to fibroblast cells [31], although other concentrations might be efficient as well. One of the limitations of this study is the use of only one concentration of each drug. Examining different concentrations of the drugs may result in protein expression and cilia length that are closer to the results obtained from healthy controls. In the current study, Gentamicin treatment was able to restore FAM161A expression weakly and partially restore co-localization with α-tubulin, but did not improve the percentage of ciliated cells. Similar findings were reported previously after Gentamicin treatment of USH2A patient-derived fibroblasts [31].

Previous studies demonstrated the read-through efficacy of Ataluren on several diseases causing PTCs, both in cellular [31,35,40] and animal models of eye disorders [29]. Furthermore, Ataluren was approved for the treatment of Duchenne muscular dystrophy (DMD) and cystic fibrosis (CF) caused by in-frame nonsense mutations in the US and has orphan drug status and conditional authorization for DMD in Europe [17]. More recently, topical application of Ataluren was shown to revert congenital tissue malformation defects in a mouse model for Aniridia [30]. In the current study, we demonstrated that Ataluren was able to restore FAM161A expression in *FAM161A*-mutated cells as well as its co-localization with α-tubulin along the microtubules (similar to the level that was observed in cells from healthy controls).

Read-through of a PTC could result in the integration of an amino acid that is different from the original one. The probability of the amino acid being integrated at the PTC site varies depending on the stop codon and the applied TRIDs [22,23,31]. The alternative amino acid might have an impact on the stability, localization and/or function of the resulting FAM161A protein. However, the series of analyses performed in the current study, including staining with FAM161A antibody showing restored FAM161A staining and co-localization of FAM161A with anti α-tubulin and functional cilia analysis, resulted in better ciliogenesis and longer cilia length in patient-derived cells treated with Ataluren.

TRIDs, and mainly Ataluren, were shown to induce read-through of nonsense mutations in genes that are expressed in the retina effectively. Different systems of drug delivery to the vitreous and the retina are currently under development and approval [41,42] and may in the future serve as an efficient platform for Ataluren delivery to photoreceptors.

In this study, we demonstrated that small molecules such as Ataluren and aminoglycosides such as Gentamicin can successfully read-through a nonsense *FAM161A* mutation resulting in functional rescue of FAM161A in human fibroblasts. This study provides further evidence that Ataluren, Gentamicin and possibly other similar compounds have the potential to restore expression and function of FAM161A in patients with the nonsense mutation p.Arg523*, which causes RP. Our results provide a proof-of-concept that PTCs in retinal genes can be treated with TRIDs.

## Figures and Tables

**Figure 1 ijms-23-03541-f001:**
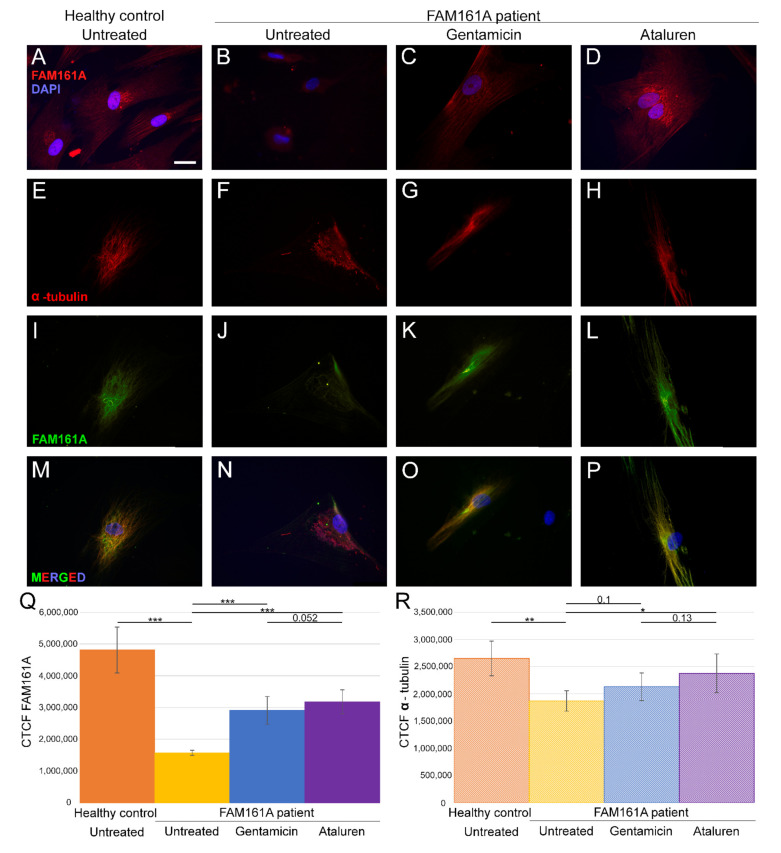
FAM161A expression and co-expression with α-tubulin in fibroblast cells. (**A**–**D**) FAM161A expression (red) after 48 h of starvation in cells from a healthy control (**A**), cells from *FAM161A* patient without treatment (**B**), following Gentamicin treatment (**C**) and Ataluren treatment (**D**). Double staining of α-tubulin (red) and FAM161A (green) in starved fibroblasts from healthy control cells (**E**,**I**,**M**), untreated fibroblasts from *FAM161A* patient (**F**,**J**,**N**) and cells from the same patient treated with Gentamicin (**G**,**K**,**O**) and Ataluren (**H**,**L**,**P**). DAPI: nucleus (blue). Scale bar 25 µm. (**Q**,**R**) Quantification of FAM161A (**Q**) and α-tubulin (**R**) expression levels in fibroblasts. Fluorescence intensities of FAM161A and α-tubulin in healthy untreated cells (orange) and untreated (yellow) Gentamicin-treated (blue) and Ataluren-treated (purple) cells from patients were measured using ImageJ, and the corrected total cell fluorescence (CTCF) was calculated (arbitrary units). The significance levels were set when *p* < 0.05 (*), *p* < 0.01 (**), *p* < 0.001 (***).

**Figure 2 ijms-23-03541-f002:**
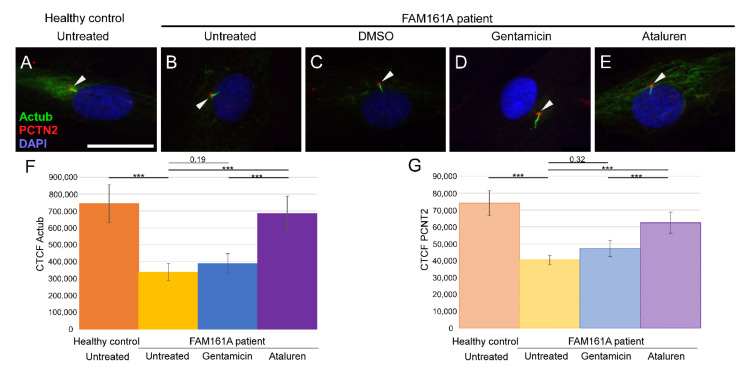
Ciliogenesis in fibroblast cells of *FAM161A* patients following TRID treatment. Immunofluorescence analysis of fibroblast cells from healthy controls (**A**) and *FAM161A* patient cells (**B**–**E**) after triggering ciliogenesis by starvation for 48 h. Antibodies against acetylated tubulin (Actub, green) marker for cilia, pericentrin (PCTN2, red) stains the basal body, and nuclei are stained by DAPI (blue). Cilia are marked with white arrows. *FAM161A* patient cells were left untreated (**B**), or treated with DMSO as control (**C**), Gentamicin (**D**) or Ataluren (**E**). Scale bar 25 µm. (**F**,**G**) Quantification of Actub (**F**) and PCNT2 (**G**) expression levels in fibroblasts. Fluorescence intensities of Actub and PCNT2 in healthy untreated cells (orange) and untreated (yellow) Gentamicin-treated (blue) and Ataluren-treated (purple) cells from patients were measured using ImageJ, and the corrected total cell fluorescence (CTCF) was calculated (arbitrary units). The significance levels were set when *p* < 0.001 (***).

**Figure 3 ijms-23-03541-f003:**
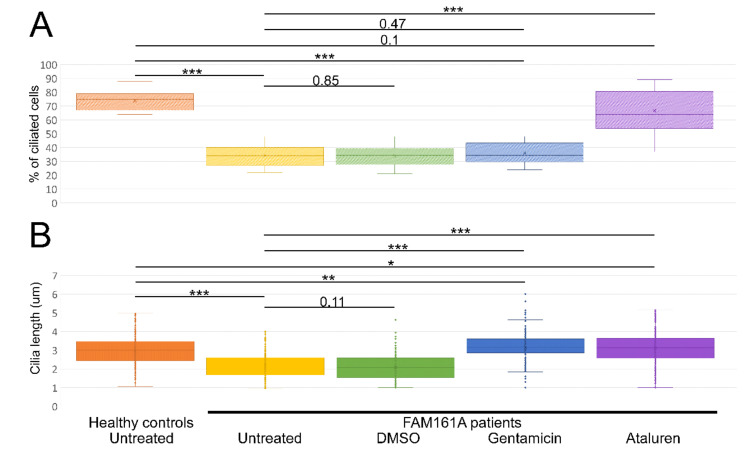
Ciliogenesis and cilia length analyses. (**A**) A boxplot displaying percentage of ciliated cells among untreated fibroblasts from five healthy controls and six *FAM161A* patients (untreated or after different treatments), obtained in three independent experiments (a total of 300 cells per each healthy control or patient were counted). For quantification of the percentage of ciliated cells, the number of DAPI stained nuclei was correlated to the number of ciliated cells (Figure 3A; *y*-axis). (**B**). Boxplot of cilia length on untreated fibroblasts from five healthy controls (*n* = 1107 cells) and six FAM161A patients (untreated or after different treatments) (*n* = 3075), obtained in three independent experiments. The ciliary length was determined based on the acetylated tubulin staining (Figure 3B; *y*-axis). The significance levels were set when *p* < 0.05 (*), *p* < 0.01 (**), *p* < 0.001 (***).

**Table 1 ijms-23-03541-t001:** Evolutionary conservation of the altered amino acid in the *FAM161A* gene. The arginine (R, marked in red) that is altered in the analyzed nonsense mutation is fully conserved through all species. Accession numbers of protein sequences are: human (NP_001188472), chimpanzee (XP_016804096), rhesus monkey (XP_028687476), cow (XP_005212882), dog (XP_038536623), rat (XP_017454620), mouse (XP_006514891) and zebrafish (XP_686612).

Species	Protein Sequence
Human	PPVPTVSSRGREQAVRRSLE
Chimpanzee	PPVPTVSSRGREQAVRRSLE
Rhesus monkey	PPVPTVSSRGREQAVRRSLE
Cow	PPTPTVSSRGREQATRRSLE
Dog	PPMPTVSSRGREQATRRSLE
Rat	PPMPTASSRGREQAIRKSLE
Mouse	PPMPTASSRGREQAIRKSLE
Zebrafish	SAKITDAAKKRQEAVRKVLE

**Table 2 ijms-23-03541-t002:** Summary of ciliogenesis and cilia length analyses. Summary of the average percentage (%) of ciliated cells and cilia length in fibroblasts from healthy controls and patients without any treatment and following different treatments.

		Untreated		DMSO		Gentamicin		Ataluren	
		% Ciliated Cells	Cilia Length	% Ciliated Cells	Cilia Length	% Ciliated Cells	Cilia Length	% Ciliated Cells	Cilia Length
Healthy controls	Average	73.8	2.95	76.91	3.05	74.16	3.11	77.33	3.01
	SEM	3.25	0.04	1.99	0.08	1.97	0.07	2.83	0.05
Patients	Average	34.27	2.13	33.83	2.06	36.11	3.14	66.61	3.1
	SEM	2.8	0.04	2.02	0.05	2.19	0.06	4.36	0.07

## Data Availability

The data presented in this study are available on request from the corresponding author. The data are not publicly available due to awkwardness in displaying the data.

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
