# Peer review of "Translational Read-Through Drugs (TRIDs) Are Able to Restore Protein Expression and Ciliogenesis in Fibroblasts of Patients with Retinitis Pigmentosa Caused by a Premature Termination Codon in *FAM161A"

_ijms, 2022, doi:10.3390/ijms23073541_

Round 1

Reviewer 1 Report

In the study by Beryozkin et al., the authors have measured and compared the numbers of cilia and cilia length in fibroblast cultures from RP patients with a missense substitution in FAM161A protein, after Gentimicin and Ataluren treatments as compared to that in untreated fibroblasts from the patients and healthy controls. The study shows promising results, however some more validation of the results is required. Here are my specific comments- 

  1. In the methods, please mention how the patients were clinically identified to be suffering from Retinitis Pigmentosa i.e using any behavioral tests by the ophthalmologists to identify RP? Were they on any drugs or vitamins for managing RP?
  2. Lines 196-199: It is not clear in how many untreated cells out of 300, was there a faint staining for tubulin, and how many such cells were seen per patient? 
  3. Lines 212-215 and Figure 1: The authors say “Ataluren treatment yielded higher FAM161A expression”. In how many cells out of 300, did they see this “higher” expression? To compare the FAM161A protein levels between different conditions, it would be helpful to do some kind of quantitation of the FAM161A protein- either by quantifying the immunolabeled cells using image anlaysis softwares such as ImageJ or using Western blotting.
  4. Lines 229-231: The statement “Both markers revealed fainter and diffuse expression in untreated mutant fibroblasts compared to controls” is not evident in the Figure 2F and G, especially for Pericentrin2. Please use representative images. Also, the authors should consider measuring total protein concentration using western blots for tubulin and pericentrin2 to further validate their findings. 
  5. Figure 2: The low magnification images especially 2A-E are not helpful, please remove them as they don’t provide any additional information. Just keep 2F-J and the cilia can be indicated by white arrowheads in 2F-J. 
  6. Figure 3: The cilia length after Ataluren treatment was significantly higher than the healthy controls. Please discuss if this holds any significance especially if Ataluren has to be used as a therapeutic drug.

Reviewer 2 Report

The authors provide an interesting manuscript on the effects of TRIDs in restoring the protein expression and ciliogenesis in fibroblasts of RP patients. The manuscript is clear and they did a nice job presenting complex findings. The authors should add a few additional details.  Further specific comments are included below.

       # it would be more informative if the authors can provide the brightfield image for Figure1.

# the fibroblasts were taken from six patients will authors be able to provide the representative Images of the six patients with brightfield images as the supplementary figure.

# line 197 and 199, authors mention fiber-like structure was not detected always. What percentage of the cell was seen with and without fibers?

Reviewer 3 Report

Accept in present form

Author Response

We thank the reviewer for the positive feedback.

Round 2

Reviewer 1 Report

All the comments have been answered satisfactorily and appropriate changes made in the manuscript.

The revised version of the manuscript is suitable for publication.